# Identification of Equine Arteritis Virus Immunodominant Epitopes Using a Peptide Microarray

**DOI:** 10.3390/v14091880

**Published:** 2022-08-26

**Authors:** Jo Mayers, David Westcott, Falko Steinbach

**Affiliations:** 1Surveillance and Laboratory Services Department, Animal Health and Veterinary Laboratories Agency, Staplake Mount, Devon EX6 8PE, UK; 2Virology Department, Animal Health and Veterinary Laboratories Agency (Weybridge), New Haw, Surrey KT15 3NB, UK; 3School of Veterinary Medicine, University of Surrey, Guildford GU2 7AL, UK

**Keywords:** ELISA, equine viral arteritis, Equine arteritis virus, peptide, micro-array

## Abstract

Using the commercially available PEPperCHIP^®^ microarray platform, a peptide microarray was developed to identify immunodominant epitopes for the detection of antibodies against Equine arteritis virus (EAV). For this purpose, the whole EAV Bucyrus sequence was used to design a total of 1250 peptides that were synthesized and spotted onto a microarray slide. A panel of 28 serum samples representing a selection of EAV strains was tested using the microarray. Of the 1250 peptides, 97 peptides (7.76%) showed reactivity with the EAV-positive samples. No single peptide was detected by all the positive serum samples. Seven peptides repeatedly showed reactivity above the cut-off and were considered to have diagnostic potential. Five of these peptides were within the immunodominant GP5 protein and two were within the replicase polyprotein regions NSP2 and NSP10, located in ORF1. The diagnostic sensitivity of the seven peptides selected was low, ranging from 5% to 55%; however, the combined diagnostic sensitivity and specificity of the seven peptides was 90% and 100%, respectively. This data demonstrate that multiple peptide sequences would be required to design a comprehensive serological test to cover the diversity of the EAV strains and the individual immune responses of horses.

## 1. Introduction

Equine arteritis virus (EAV) is the causative agent of equine viral arteritis (EVA), a respiratory and reproductive disease of horses [1] EAV only infects equids, causing a wide range of clinical signs that cannot be differentiated clinically from a number of other respiratory and reproductive equine diseases. The most common of these are equine influenza, equine herpes virus 1 and 4, infection with equine rhinitis A and B viruses, equine encephalosis, and streptococcal infections [2]. The disease is principally characterized by fever, lethargy, conjunctivitis (“pink eye”), anorexia, limb oedema, abortion, or neonatal foal death. Affected adult horses make a complete recovery, but a percentage of infected stallions may become long-term carriers [3,4].

EAV, formally now classified as *Alphaarterivirus equid*, is a spherical, enveloped, single-stranded, positive-sense RNA virus and is the sole member of the genus *Alphaarterivirus* in the subfamily *Equarterivirinae* of the *Arteriviridae*. This family also includes porcine reproductive and respiratory syndrome virus (PRRSV), lactate dehydrogenase-elevating virus (LDV) and many other members [5]. Arterivirus infections are species specific; however, they share biological and molecular properties, including their virion morphology, replication strategy, and their ability to establish persistent infections [6]. The EAV virion consists of four structural proteins, a phosphorylated nucleocapsid protein (N) and three membrane proteins, M, Gs (GP2), and GL (GP5). The genome of EAV has a length of 12.7 kb and contains seven open reading frames (ORFs) [7]. ORFs 1a and 1b represent seventy-five percent of the viral genome and code for replicase and polymerase activities [8]. ORFs 2–7 are nested and code for the structural proteins of the virus. ORF 2 encodes the Gs protein. ORF 3 and 4 are uncharacterized but are predicted to have non-structural functions. ORF 5 encodes the GL protein an immunodominant component, which is known to express the known neutralizing determinants. ORF 6 encodes the M protein and ORF 7 the N protein [9]. Although there is only one serotype of the EAV, variation exists between field strains of the virus in both their neutralizing and virulence properties [10]. The phylogenetic classification of EAV has changed over time and, while formerly two major groups (group I, mainly comprising viruses from North America, and group II viruses, mainly from Europe) were identified, a more recent refinement means that we currently differentiate between seven genotypes (A–G), no longer linked to geographic regions [11].

The main methods of controlling EAV are vaccination and the establishment of freedom from infection for (frozen) semen before breeding activities commence. Animals that are sero-positive can be re-tested to confirm that they are no longer infectious, indicated by declining antibody levels [12] (HBLB, 2022). Previous serological and clinical studies have indicated that EAV is widely distributed in equine populations around the world. Therefore, many countries carry out testing prior to the arrival of horses [2,11].

The diagnosis of EAV infection is based on the virus isolation, the detection of viral nucleic acid or viral antigen, or the demonstration of a specific antibody response. At present, the serum neutralization test (SNT) is the principal serological assay used to detect EAV infection and continues to be the World Organization for Animal Health (OIE) recommended test for EVA in individual horses [13]. Although the SNT is currently the most specific diagnostic test available, it is expensive, labor intensive, time-consuming to perform, and carries an inherent risk of contamination. Therefore, further re-search is warranted to provide a more rapid and sensitive serological diagnostic test for EAV.

Advances in protein technology enable screening to be undertaken to identify suitable targets for the development of new diagnostic tests. The PEPperCHIP^®^ peptide microarray platform is a tool designed for the identification of antibody profiling/epitope mapping. The surfaces of these custom-made microarrays are coated with an amino-modified poly(ethylene glycol)methacrylate (PEGMA), which permits a high resolution of amino acid particles to be deposited onto the surface of the slide [14]. The fundamental basis of the assay is the same as for a serological ELISA: a series of incubation and wash steps result in the capture of antibodies and the subsequent detection by a labelled anti-species secondary antibody [15].

The aim of the present study was to identify the key immunodominant epitopes of EAV using a peptide microarray. The identification of peptide sequences commonly detected by sera from EAV-infected horses could provide new opportunities for establishing a serological test, such as a next-generation ELISA.

## 2. Materials and Methods

### 2.1. Custom-Made EAV PEPperCHIP^®^ Microarray

The Equine arteritis virus Bucyrus nucleic acid sequence (GenBank accession number X53459) was used to generate peptide sequences across all open reading frames. The EAV proteins were arranged into six hundred and twenty-five 15-mer peptides with a shift of seven amino acids. The peptide microarray was printed with each slide containing five identical array replicates (sub-arrays), and each sub-array was flanked (top and bottom) with a row of 72 FLAG peptides (DYKDDDDK), a synthetic octapeptide, and 68 haemagglutinin (HA) peptides (YPYDVPDYAG) from the H3 influenza strain as the control peptides. The surface of the microarray glass slide was coated with an amino-modified poly(ethylene glycol)methacrylate (PEGMA), enabling amino acid particles to be deposited onto the surface of the slide.

### 2.2. Determination of the Peptide Sequences

A panel of 28 serum samples was selected that were previously analyzed by SNT, as described [13,16], to determine their serological status. The serum sample titers, as determined by the SNT, ranged from 1/4 to 1/4096 (Table 1). Sample 1 represents a positive control serum that is used routinely at the Animal and Plant Health Agency (APHA) in the SNT, and it was tested on all microarray slides to act as a normalizer. All tested sera related to EVA cases where the EAV strains had been (partially) sequenced at the time. Additional serum samples were specifically chosen, as they had been incorrectly identified previously using a commercial EVA ELISA kit. The samples were isolated between 1993 and 2012 from a range of horse breeds and included animals of different age groups and both sexes.

### 2.3. Microarray Protocol

To perform the peptide microarray, each glass slide was placed into a microarray slide holder, which can hold up to a maximum of three microarray slides, creating a well-based platform for each of the five individual ‘sub-arrays.’ Each individual well was pre-absorbed with 200 μL of standard buffer (phosphate buffered saline pH 7.4, 0.05% Tween 20) and incubated for 10 min at room temperature on a shaker. The standard buffer was removed before 200 μL of blocking buffer (phosphate buffered saline pH 7.4, 0.05% Tween 20 and 1% bovine serum albumin) was added to each well and incubated on a shaker for one hour at room temperature. The slides were washed three times by adding 500 μL of standard buffer and placing on the shaker for 10 s at room temperature. Secondary antibody polyclonal goat anti-mouse immunoglobulin labelled with Cy5b and polyclonal goat anti-horse immunoglobulin labelled with Cy3b were diluted at 1/500 in staining buffer (phosphate buffered saline pH 7.4, 0.05% Tween 20, and 0.1% bovine serum albumin) before adding 200 μL to each well and incubation for one hour on a shaker at room temperature. The slides were washed as described above, then rinsed with molecular grade water, before drying with a stream of air and reading on the microarray scanner to determine any non-specific binding. The slides were returned to the slide holder and blocked as described previously. Serum samples were diluted at 1/1000 in staining buffer and 200 μL was added to each well and they were incubated overnight on a shaker at ±4 °C. The arrays were washed as described above, and 200 μL of secondary antibodies diluted at 1/500 in staining buffer was added to each well before incubation for 1 h on a shaker at room temperature. The slides were washed as described above, and dried using a stream of air before being read on the microarray scanner to determine the sample signal intensity. The slides were returned to the slide holder and incubated with 200 μL of internal controls (PEPperPRINT—mouse anti-FLAG Cy3, mouse anti-HA Cy5 labelling kit), then diluted at 1/1000 in staining buffer for one hour on a shaker at room temperature. The slides were washed as described above, then rinsed with molecular grade water, before drying with a stream of air. Finally, the slides were read a third time using the microarray scanner (Agilent, Stockport, UK) to obtain the signal intensities of the internal controls.

### 2.4. Microarray Data Extraction and Analysis

Data were extracted from the microarray images using the Agilent feature extraction software (version 10.7, Agilent, Santa Clara, CA, USA). The fluorescent intensities were acquired using Axon GenePix Pro 6.0 software (Molecular Devices, San Jose, CA, USA) and the raw data were transferred to Microsoft Excel. Outlier measurements were detected using a Chi-square test, as performed in the outlier package in R (http://www.r-project.org/ (accessed on 15 July 2022) version 2.15.1f. The significance level of the outlier detection was set at *p* < 0.0000001 and used for further analysis.

### 2.5. Synthetic Peptides

Selected peptide sequences were subsequently synthesized by PEPperPRINT according to their standard protocols and supplied as high-performance liquid chromatography (HPLC)-purified lyophilized powder. The peptides were reconstituted to a concentration of 1 mg/mL in 10% Dimethylformanide (DMF) with molecular grade water using a sonicator at room temperature until the peptides had dissolved.

### 2.6. Evaluation and Optimization of the Synthetic Peptides in an ELISA Format

The enzyme-linked immunosorbent assay (ELISA) was performed in a 96-well micro-titer plate (MaxiSorb, NUNC, Roskilde, Denmark) and coated with 100 µL of synthetic peptide diluted in 0.05 M carbonate buffer pH 9.6 (Sigma, Gillingham, UK) overnight at ±4 °C. Optimal coating concentrations were evaluated by titrating the peptides. The plates were washed three times with 200 µL wash buffer (phosphate-buffered saline (PBS) containing 0.05% Tween 20) and tapped out onto absorbent towels to remove any excess. The wells were blocked with 100µL of serum/conjugate diluent (0.5 M phosphate buffer with 0.5 M NaCl, 1 mM EDTA, 1% polyvinyl pyrrolidone 40 and 0.05% Tween 20, at pH 7.2) and incubated at room temperature for one hour. Following the incubation, the plates were washed three times, as described above. The serum samples were diluted at 1/100 in serum/conjugate diluent and 100 µL was transferred to each well and they were incubated for one hour at room temperature. The plates were washed as described above, and 100 µL of goat anti-horse conjugated to horse-radish peroxidase (HRP)b was diluted at a range of concentrations in the serum/conjugate diluent before adding to each well, and they were incubated for 1 h at room temperature. Finally, the plates were washed, as described above, and 100 µL of 3,3′,5,5′-Tetramethylbenzidine (TMB) substrate was added to each well and incubated in a dark at room temperature for a minimum of 10 min. Following visual inspection, the enzymatic reaction was stopped by adding 50 µL of stop solution (2 M sulfuric acid). The optical density of each well was measured using an ELISA plate reader at 450 nm.

## 3. Results

In total, 1250 peptide sequences were generated through the computational analysis, synthesized, and spotted onto the PEPperCHIP^®^ microarray platform. An additional 72 FLAG and 68 HA proteins were printed onto the microarray as a form of internal control. The HA protein had the additional benefit of detecting any horse vaccinated against equine influenza (EI), as the H3 subtype is generally incorporated into the EI vaccine, therefore acting as an internal positive control [17].

There was good agreement between the microarray and the SNT results. However, three serum samples that were positive according to the SNT (sample 11, 13 and 20) were not detected by the microarray. Subsequent analysis discovered that the recommended dilution for the commercial ELISA positive control (sample 20) in an ELISA format at a 1/10 dilution was compared to the 1/1000 dilution that was used in the microarray protocol. The dilution factor increase may have contributed to the fact that it was not detected, and it was retested at a 1/10 dilution, leading to the discovery that it was detecting peptides two and four (Table 1). A neutralizing monoclonal antibody (mAb) that is specific against EAV (sample 21) was also included in the panel to see if it could detect a linear peptide. The results demonstrate that the mAb was not detected by the microarray and neither a conformational epitope nor a linear epitope was detected that was not represented in this array. Therefore, the results from the mAb were not included in any further downstream analysis. Samples negative according to the SNT showed no detectable signals on the microarray.

From a total of 1250 peptides, 97 peptides (7.76%) showed reactivity above the cut-off, as determined by the *p*-value (*p* < 0.0000001). The peptides that showed reactivity above the cut-off were analyzed manually to identify which combinations of peptides would successfully capture all known EAV variants. Based on the results, seven peptides repeatedly showing reactivity above the cut-off were considered to have diagnostic potential (Figure 1). These seven peptides identified were subjected first to a protein blast search (http://blast.ncbi.nlm.nih.gov/Blast.cgi (accessed on 13 July 2022)) to confirm that they were homologous to EAV only (data not shown).

As observed in Table 1, no single peptide was detected by all the positive serum samples. Five peptides were within the GP5 protein region, three of which were identified with a wider region of antigenicity. The remaining two peptides were within the replicase polyprotein regions NSP2 and NSP10, located in ORF1 (see Table 2 for the peptide sequences and locations).

The diagnostic sensitivity and specificity of each individual peptide were calculated using a 2 × 2 box analysis (Table 3). While the diagnostic specificity of these seven peptides was 100% (according to the reactivity of the sera in the SNT) their diagnostic sensitivity varied considerably. The diagnostic sensitivity of the seven selected peptides was low, ranging from 5% to 55%. Peptide seven had the lowest diagnostic sensitivity out of all the selected peptide candidates. However, this peptide was specifically selected because of its ability to detect the EAV vaccine strain (Artervac). The combined diagnostic sensitivity and specificity of the seven peptides (excluding the mAb) was 90% and 100%, respectively.

The peptide sequences that showed reactivity across the selected horse sera were identified as potential candidates for the evaluation of their application in an ELISA-based format. The two most reactive peptides (peptides 1 and 2) were synthesized and further evaluated as synthetic peptides. The optimal coating concentrations of the synthetic peptides were determined by coating them at various concentrations, combined with a selection of secondary antibody dilutions in the form of a checker-board titration. The ELISA results demonstrate that it was possible to passively absorb the synthetic peptides on standard ELISA plates and to detect the positive control serum (data not shown).

## 4. Discussion

A number of EAV ELISAs have been developed over the years [7,18,19,20,21,22], using various biological agents as capture antigens in an attempt to provide a sensitive serological assay equivalent to the SNT. However, none of the currently available ELISAs have been sufficient to provide a diagnostic test for individual animals, and both false positive and false negative reactions have been reported as occurring frequently. ELISA tests using crude viral lysates have been reported to lack sensitivity due to their high background reactivity with equine sera [20]. In contrast, the recombinant ORF5 protein expressed in the baculovirus system, which is used as an antigen, differs antigenically from the same protein in its virion-associated form [7]. None of the ELISAs have shown equivalent diagnostic sensitivity and specificity to the SNT; therefore, it is considered the ‘gold standard’ for detecting antibodies to EAV and the prescribed test for international trade [13]. However, it is widely accepted that sera tested on the EAV SNT from horses vaccinated against equine herpes virus 1 (EHV1) can cause serum cytotoxicity, causing difficulties in test interpretation [23].

EAV, like all other Arteriviruses, are rapidly evolving organisms, which at least in part explains the lack of specificity in the existing ELISA formats. The fact that the antibody response of the individual horse differs depending on the host genetics, time of infection, EAV strain, and the duration of the infection contributes to the false identification and provides further explanation for the difficulties observed in developing a serological assay. Previous studies using the GL/ORF5 protein [22,24] have indicated that the most promising method for diagnostic purposes would be achieved with synthetic peptides. The use of peptides as antigens in serological assays has increased in recent years [24,25] due to significant benefits, such as the stability and reproducibility they offer [15]. They have been applied successfully for the diagnosis of many pathogens, including the related PRRSV as a model [26,27].

This study evaluated the complete Bucyrus proteome for the identification of suitable antigenic epitopes in order to identify a set of targets for subsequent assays. The selected approach consisted of assessing a diverse range of equine sera from horses infected with different strains based on sequencing data, representing strains from both the European and USA genotypes, as well as vaccine and negative sera.

The microarray results produced large amounts of data that are difficult to analyze and interpret. To overcome these potential problems, a computer script was written in “R” that was designed to differentiate between the positive and negative results using a *p*-value cut-off. Multiple *p*-value cut-offs were evaluated in this study to determine the best cut-off value for implementation, as decreasing the *p*-value increases the confidence in the data and minimizes the potential of false positive results. However, the *p*-value used in this study was deliberately selected at *p* < 0.0000001 to eliminate the potential of false positive results. This approach involves the risk that relevant epitopes were missed or excluded from the further downstream analysis.

As expected, the linear peptides harbored a series of immunodominant epitopes to EAV, with 7.76% showing a positive reaction to at least one of the sera. This correlates with other studies, which estimate that approximately 10% of antibodies detected are against linear epitopes [28]. While seven peptides were chosen for progression to further analysis, there are many more peptides that were identified with diagnostic potential, which could also be used in combination with the peptides discussed in this study. Among the peptides identified in this study, the GP5 protein was found to be the most immunodominant protein. Peptide 2 was the most reactive peptide, with a diagnostic sensitivity of 55%, although on its own did not meet the sensitivity required for antigens applied in routine diagnosis.

From the panel of selected sera, samples 11 and 13 were not detected by any of the selected peptides candidates, which may be due to a number of factors. Firstly, these samples were not undetected due to the stringent *p*-value that was used in this study, as no signal above the background noise was observed. Therefore, further investigation and retesting will need to be carried out to determine the possible cause.

Evidence from several different investigations [18,24,29,30] has illustrated the importance of the GP5 as an antigen, which correlates with the results of this study. However, the GP5 protein alone is not sufficient to detect all known variants. Evaluating peptide sequences that represent the whole of the virus has enabled the capture of all the potential immunodominant sites. The results presented here support the concept that a mixture of peptides is required to ensure sufficient coverage. As expected, a combination of several peptides increased the sensitivity, which is consistent with reports from other diagnostic peptide studies [24,25]. One solution would be to combine three of the overlapping 15-amino acid peptides together to form the basis of a single novel artificial antigen. However, researchers must test how these peptides will react in another format and if any secondary folding between the peptides will occur, inhibiting the binding of the antibodies to the specific epitope regions. However, it has been suggested that a maximum length of 30 amino acids is recommended for synthetic peptides before sensitivity is lost [25], demonstrating that the choice and design of the peptide is of major importance for a successful peptide ELISA.

As the data demonstrate, a single peptide is not sufficient to cover the whole diverse range of the EAV. Some of the variability observed in the antibody response of the individual horse infected with EAV may be attributed to the different field strains of the virus and/or the interval following infection [7,31]. Heterogeneous pathogens such as EAV have been shown to induce a dynamic range of responses against multiple epitopes, with each virus variant having several epitopes which can stimulate different immune responses. The antigenic escape of one epitope can lead to the change in the genetic structure of the virus quasispecies population and the specificity of the immune response of the host, therefore altering the immunodominant response to another epitope [32]. This provides further supporting evidence that several peptides are required to cover all the antigenic variations of EAV.

Developing assays with synthetic peptides has the added value of consistency in terms of the reproducibility, purity and the versatility of modified or labelled synthetic peptides [15], providing opportunities for the use of alternative diagnostic detection systems, such as multiplex-based technologies [33]. The method described in this study also offers the potential for the peptide identification of other pathogens that require handling in high-containment facilities or where limited material is available. However, one of the shortcomings of the use of synthetic peptides is the potential loss of epitopes, resulting in the destruction of the antibody binding site. Although synthetic peptides offer a promising alternative to crude antigens, these results should be seen as an initial assessment. A more thorough evaluation is required to determine whether these peptides perform comparably to the SNT. We may find that testing the selected peptide candidates in an ELISA format increases the combined diagnostic sensitivity of the peptides, due to the larger surface area of the ELISA well compared with the microarray, or even reduces the number of peptides required to cover all EAV strains and immunological responses of the host.

In summary, a peptide microarray is a useful tool for the identification of suitable peptides for the detection of antibodies to EVA. The results from this study have identified several key peptide sequences from the immunodominant GP5 protein and replicase polyprotein region, which could be suitable for the development of a new serological test. The data demonstrate that one peptide sequence is not sufficient to cover all the EAV strains; therefore, a selection of peptide sequences have been identified as potential candidates. Based on the results obtained in this study, a peptide ELISA offers a promising approach as an alternative to the SNT for the detection of antibodies to EAV. However, at present, the SNT remains the most sensitive test for detecting EAV-specific antibodies in equine serum. Further development and optimization are required to determine whether the peptide candidates identified in this study offer the diagnostic sensitivity and specificity required for the diagnosis of EAV infection in individual animals.

## Figures and Tables

**Figure 1 viruses-14-01880-f001:**
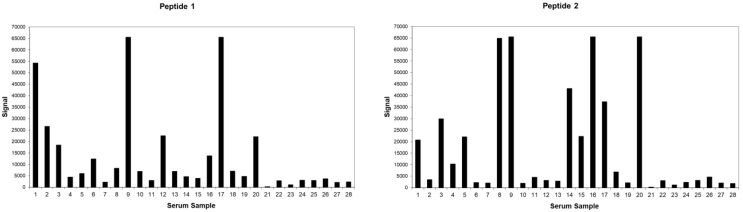
Relative signal intensity of EAV peptides when tested against horse sera.

**Table 1 viruses-14-01880-t001:** Comparison of the serological response of serum from EAV infected horses to seven EAV peptides on the microarray.

Sample Number	Strain	SNT	Peptide 1	Peptide 2	Peptide 3	Peptide 4	Peptide 5	Peptide 6	Peptide 7
**1**	USA	3200	+	+	−	+	+	−	−
**2**	USA	1024	+	−	−	−	−	−	−
**3**	USA	512	+	+	+	−	+	−	−
**4**	USA	1536	−	+	−	−	+	+	−
**5**	USA	4	+	+	−	−	−	−	−
**6**	USA	1024	+	−	+	+	−	−	−
**7**	USA	32	−	−	−	−	−	+	−
**8**	USA	128	−	+	−	+	−	−	−
**9**	USA	1536	+	+	−	+	+	−	−
**10**	USA	128	+	−	−	−	−	−	−
**11**	USA	4096	−	−	−	−	−	−	−
**12**	USA	128	+	−	+	+	−	−	−
**13**	USA	2048	−	−	−	−	−	−	−
**14**	European	>2048	−	+	+	+	−	+	−
**15**	European	2048	−	+	−	−	−	−	−
**16**	European	1024	−	+	−	−	−	−	−
**17**	European	6144	+	+	−	−	−	−	−
**18**	EAV Vaccine	1024	−	−	−	−	−	−	+
**19**	EAV Vaccine	1024	−	−	+	−	−	−	−
**20**	ELISA kit +	192	−	+	−	+	−	−	−
**21**	mAb	3072	−	−	−	−	−	−	−
**22**	Negative	<4	−	−	−	−	−	−	−
**23**	Negative	<4	−	−	−	−	−	−	−
**24**	Negative	<4	−	−	−	−	−	−	−
**25**	Negative	<4	−	−	−	−	−	−	−
**26**	Negative	<4	−	−	−	−	−	−	−
**27**	Negative	<4	−	−	−	−	−	−	−
**28**	Negative	N/A	−	−	−	−	−	−	−

Abbreviations: SNT = serum neutralization test; USA = strain within EAV North America group I lineage; European = strain within EAV European group II lineage; EAV = Equine arteritis virus; mAB = monoclonal antibody; + = positive; − = negative.

**Table 2 viruses-14-01880-t002:** EVA peptide sequences and locations.

Peptide	Protein Sequence	Region
**1**	DCNDTYAVPVAEVLE	GP5
**2**	EQAHGPYSVLFDDMP	GP5
**3**	TDRGVIANLLRYDEH	GP5
**4**	VPVAEVLEQAHGPYS	GP5
**5**	VQRARSVGDVLVQAL	NSP2
**6**	AEGKVYPVDPGLPVA	GP5
**7**	VIHSYPNCGPACGWE	NSP10

Abbreviations: A = alanine; C = cysteine; D = aspartic acid; E = glutamic acid; G = glycine; H = histidine; I = isoleucine; L = leucine; M = methionine; N = asparagine; P = proline; Q = glutamine; R = arginine; S = serine; T = threonine; V = valine; W = tryptophan; Y = tyrosine.

**Table 3 viruses-14-01880-t003:** Diagnostic sensitivity and specificity of the seven EAV peptides that were selected based on their performance on the peptide microarray (as determined by the cut-off (*p* < 0.0000001)).

	Peptide 1	Peptide 2	Peptide 3	Peptide 4	Peptide 5	Peptide 6	Peptide 7
Sensitivity (%)	45	55	25	35	20	15	5
Specificity (%)	100	100	100	100	100	100	100

## Data Availability

The data presented in this study are available on request from the corresponding author.

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
