# Peer review of "Identification of Equine Arteritis Virus Immunodominant Epitopes Using a Peptide Microarray"

_viruses, 2022, doi:10.3390/v14091880_

Round 1
Reviewer 1 Report
The manuscript entitled "Identification of Equine Arteritis Virus Immunodominant Epitopes Using a Peptide Microarray" by Mayers et al. describes the development and initial validation of a commercially available peptide microarray to test equine serum samples for the presence of EVA specific antibodies. The manuscript is well-written and the results are clearly presented. As the authors state, the development of a diagnostic assay facilitating serological detection of EVA specific antibodies is of some importance to the equine industry. However, there are a few items that may warrant attention:
1. Introduction, line 34: Please replace the term "depression" with "lethargy".
2. Introduction, line 34-35: Please add "neonatal foal death" to the disease description.
3. Introduction, line 62: While serological testing has importance for horse movement, virological testing is central to demonstrate freedom of infection for (frozen) semen.
4. Introduction, line 66: Please change "health checks" to "testing".
5. Materials and Methods: Please provide additional information on the serum samples used. What time frame do these samples span (i.e., would these samples represent currently circulating viruses?).
6. Results/Discussion: Please provide additional information on the results of sample 4. Unless mistaken, the sample constitutes a EHV vaccine control, but was considered to show a positive reaction to peptides 1 and 2. Please explain.
7. Figure 1: Please add a line that indicates the signal cut-off in the various graphs.
8. None of the peptides showed a high diagnostic sensitivity (although when combined their diagnostic performance improved). Could future viral evolution further impact the diagnostic sensitivity of these peptides? Also, were any patters of reactivity detected, based on the specific serum samples included (e.g., geographic location of the infected horse, year of sample collection, etc.)?
9. Discussion: Overall, the discussion is somewhat lengthy. Maybe it could be condensed to improve readability.
Author Response
Response to reviewer:
Identification of equine arteritis virus immunodominant epitopes using a peptide microarray: Jo Mayers, David Westcott and Falko Steinbach.
The authors would like to thank the reviewer for their comments and suggestions, and provide their responses as follows:
Response to Reviewer 1
- Introduction, line 34: Please replace the term "depression" with "lethargy".
Response: Revision has been made (line 34).
- Introduction, line 34-35: Please add "neonatal foal death" to the disease description.
Response: Revision has been made (line 35).
- Introduction, line 62: While serological testing has importance for horse movement, virological testing is central to demonstrate freedom of infection for (frozen) semen.
Response: The sentence has been revised to include frozen semen (line 62).
- Introduction, line 66: Please change "health checks" to "testing".
Response: Revision has been made (line 66).
- Materials and Methods: Please provide additional information on the serum samples used. What time frame do these samples span (i.e., would these samples represent currently circulating viruses?).
Response: Revision has been made (line 110).
- Results/Discussion: Please provide additional information on the results of sample 4. Unless mistaken, the sample constitutes a EHV vaccine control, but was considered to show a positive reaction to peptides 1 and 2. Please explain.
Response: The authors would like to thank the reviewer for spotting this transcription error and that it should have stated two EAV vaccine controls. The authors have updated the table and text to reflect this error as well as reformatting the table to group the positive and negative samples together to improve the format.
- Figure 1: Please add a line that indicates the signal cut-off in the various graphs.
Response: The cut-off used for each of the peptides was based on reactivity as determined by a very stringent P-value (P<0.0000001) to ensure a high specificity e.g. no false positives. As the cut-off is calculated by the statistical algorithm based on the signal intensity of each assay the cut-off will vary between runs and therefore there a single cut-off value cannot be included in the figures.
- None of the peptides showed a high diagnostic sensitivity (although when combined their diagnostic performance improved). Could future viral evolution further impact the diagnostic sensitivity of these peptides? Also, were any patters of reactivity detected, based on the specific serum samples included (e.g., geographic location of the infected horse, year of sample collection, etc.)?
Response: As with current serological tests for EAV, a single epitope/strain does not offer 100% diagnostic sensitivity. Therefore, this manuscript is trying to offer an alternative and flexible assay where a selection of capture antigens (peptides) is used to improve the serological detection of EAV. The use of peptides that can be readily and cheaply synthesised can therefore be easily changed/included to reflect current strains (with appropriate validation to ensure diagnostic sensitivity and specificity is not lost.
- Discussion: Overall, the discussion is somewhat lengthy. Maybe it could be condensed to improve readability.
Response: The author has reviewed the discussion and has shortened where possible as requested.
Reviewer 2 Report
Dear Authors, your manuscript is written very well. Please correct the marked words!

Author Response
Response to reviewer:
Identification of equine arteritis virus immunodominant epitopes using a peptide microarray: Jo Mayers, David Westcott and Falko Steinbach.
The authors would like to thank the reviewer for their comments and suggestions, and provide their responses as follows:
Reviewer 2
- Remove hyphen from Determined on line 276.
Response: Revision has been made (line 258).
- Remove hyphen from Secondary on line 277.
Response: Revision has been made (line 259).